# Validation of visual estimation of neonatal jaundice in low-income and middle-income countries: a multicentre observational cohort study

Gary L Darmstadt [ID] ,[1] Davidson H Hamer [ID] ,[2,3] John B Carlin,[4] Prakash M Jeena,[5] Eduardo Mazzi,[6] Anil Narang,[7] A K Deorari,[8] Emmanuel Addo-Yobo,[9] MAK Azad Chowdhury,[10] Praveen Kumar [ID] ,[7] Yaw Abu-Sarkodie,[11] Kojo Yeboah-Antwi [ID] ,[2,11] Pallab Ray,[7] Andres E Bartos,[12] Samir K Saha,[13] Eric Foote,[1] Rajiv Bahl [ID] ,[14] Martin W Weber[15]

For numbered affiliations see end of article.

**Correspondence to**
Dr Gary L Darmstadt;
gdarmsta@stanford.edu

## ABSTRACT

**Objective** Determine the sensitivity and specificity of neonatal jaundice visual estimation by primary healthcare workers (PHWs) and physicians as predictors of hyperbilirubinaemia.

**Design** Multicentre observational cohort study.

**Setting** Hospitals in Chandigarh and Delhi, India; Dhaka, Bangladesh; Durban, South Africa; Kumasi, Ghana; La Paz, Bolivia.

**Participants** Neonates aged 1–20 days (n=2642) who presented to hospitals for evaluation of acute illness. Infants referred for any reason from another health facility or those needing immediate cardiopulmonary resuscitation were excluded.

**Outcome measures** Infants were evaluated for distribution (head, trunk, distal extremities) and degree (mild, moderate, severe) of jaundice by PHWs and physicians. Serum bilirubin level was determined for infants with jaundice, and analyses of sensitivity and specificity of visual estimations of jaundice used bilirubin thresholds of >260 µmol/L (need for phototherapy) and >340 µmol/L (need for emergency intervention in at-risk and preterm babies).

**Results** 1241 (47.0%) neonates had jaundice. High sensitivity for detecting neonates with serum bilirubin >340 µmol/L was found for 'any jaundice of the distal extremities (palms or soles) OR deep jaundice of the trunk or head' for both PHWs (89%–100%) and physicians (81%–100%) across study sites; specificity was more variable. 'Any jaundice of the distal extremities' identified by PHWs and physicians had sensitivity of 71%–100% and specificity of 55%–95%, excluding La Paz. For the bilirubin threshold >260 µmol/L, 'any jaundice of the distal extremities OR deep jaundice of the trunk or head' had the highest sensitivity across sites (PHWs: 58%–93%, physicians: 55%–98%).

**Conclusions** In settings where serum bilirubin cannot be measured, neonates with any jaundice on the distal extremities should be referred to a hospital for evaluation and management, where delays in serum bilirubin measurement and appropriate treatment are anticipated following referral, the higher sensitivity sign, any jaundice

## Strengths and limitations of this study

► This study was designed to address the evidence gap from low-income to middle-income countries on the reliability of clinical signs to screen newborn infants for jaundice in the first 3 weeks after delivery to determine the need for facility-based care and is the largest evaluation reported to date, to our knowledge, to compare clinical assessments for jaundice in newborn infants by primary healthcare workers and physicians.

► The findings have the potential to be generalised to large populations in low-income and middle-income countries and to influence WHO guidelines.

► We did not objectively assess the depth of jaundice, and 22% of infants with some degree of jaundice did not have laboratory work done.

► Due to the characteristics of the larger study within which this assessment was embedded, there was likely participant selection bias towards infants who were considered unwell. Thus, sensitivity and specificity may be different in other populations, in particular, the general neonatal population not restricted to those considered unwell.

► This study did not include sufficient numbers of preterm infants to allow for independent assessment of the utility of jaundice detection in this subgroup of infants who are especially vulnerable to morbidity and mortality from hyperbilirubinaemia.

on the distal extremities or deep jaundice of the trunk or head, may be preferred.

## INTRODUCTION

Jaundice is among the most common neonatal problems, particularly in the first week.[1] Delayed detection and therapy place newborn infants, especially those born preterm, at risk for bilirubin encephalopathy and kernicterus, leading to neurodevelopmental impairment,

disability and potentially death.[2–5] Globally, in 2010, an estimated 481 000 newborn infants were at risk of extreme hyperbilirubinaemia, and of these, 24% were at risk of neonatal death, 13% for kernicterus and 11% for stillbirth due to severe haemolytic disease manifest in utero as progressive anaemia and hypoalbuminaemia, leading to hydrops fetalis.[6]

Prevention of morbidity and mortality due to hyperbilirubinaemia in low and middle-income countries (LMIC) could markedly improve if primary healthcare workers (PHWs) were able to accurately identify and appropriately refer newborn infants with clinically significant jaundice. The Integrated Management of Childhood Illness (IMCI) strategy to identify young infants with severe illness needing urgent referral level care was revised in 2008 based on the results of the Young Infant Clinical Signs Study (YICSS)[1] and was further validated in a community-based setting in Bangladesh.[7] In 2014, IMCI included jaundice of palms and soles as a need for urgent referral.[8] Approximately, three-fourths of countries have included jaundice of palms and soles in their use of IMCI, based largely on expert opinion.[9] Jaundice on the day of childbirth is always a pathological sign that requires referral to hospital for evaluation and management. Also, jaundice persisting or first presenting more than 3 weeks after delivery is generally pathological, and these infants also need referral for evaluation. A major gap in IMCI guidelines is the need to validate clinical signs of jaundice in newborn infants aged 1–20 days, which signal urgent referral for hospital-level care.

Several studies have examined the correlation between identification of jaundice in newborn infants by experienced physicians and serum bilirubin levels.[10–16] The extent of peripheral spread of jaundice has been advocated as a useful clinical sign for identifying infants with high levels of serum bilirubin needing intervention.[10–12 14–17] However, others have questioned the ability of healthcare providers, including physicians and nurses, to diagnose clinical jaundice,[18 19] and data are lacking on the ability of PHWs in LMIC[20] settings to identify newborn infants with clinically significant jaundice requiring referral for care. Point-of-care devices increasingly show promise as an alternative approach for identifying hyperbilirubinaemia but are not widely available.[21–23]

In this multicentre study nested in the YICSS,[1] we determined the sensitivity and specificity of a range of definitions of clinical jaundice, according to distribution and severity of skin staining as assessed by PHWs and by physicians, and the agreement between their clinical assessments, for predicting the presence of serum hyperbilirubinaemia in neonates in the first 3 weeks.

## METHODS
### Study locations and patient selection
Each of the six study locations over three continents (table 1) collected data for a consecutive 12-month period during 2003 to 2005, as described previously.[24–27] Young infants brought to the hospital for any type of acute illness were included in the main study, if they were <60 days of age and their parent or guardian provided informed consent, as described previously; the participant group for the current study was a sample of convenience based on enrolments in the main study.[1] Recruitment was stratified into two age categories—0–6 and 7–59 days—to ensure adequate sampling of neonates in the first week, when

**Table 1** Methods of bilirubin detection and quality assurance by study location, Young Infant Clinical Signs Study Group

| Study location | Method for bilirubin measurement | Quality control measures |
|---|---|---|
| (1) Dhaka Shishu Hospital, Dhaka, Bangladesh | Dimethylsulfoxide (Chronolab AG, Zug, Switzerland) colorimetric method using a Humalyzer—2000 (Human Gesellschatt for Biochemica and Diagnostic mbH, Germany) | Quality control was done daily using calibration and quality control serum from the manufacturer |
| (2) All India Institute of Medical Sciences and Safdarjung Hospital, Delhi, India; (3) Postgraduate Institute for Medical Education and Research and General Hospital, Sector 16, Chandigarh, India | Spectrophotometer (BIL-100, Cosmo Medical Co., Ltd., Seoul, Korea) with built-in auto-calibration | Bilirubin standards were used periodically (every 4–5 months) to cross-check the calibration |
| (4) King Edward VIII Hospital, Durban, South Africa | Colorimetric method using a Roche Modular P-800 spectrophotometer (Roche Modular 800 (Roche Diagnostics, Basel, Switzerland) | The machine was manually calibrated whenever a new kit was inserted or when the machine sent out an automated request for calibration, approximately once weekly |
| (5) Komfo Anokye Teaching Hospital, Kumasi, Ghana | Automated colorimetric method using a spectrophotometric autoanalyser (Atac-8000, Elan Diagnostics, Smithfield, RI, USA) | The machine was calibrated daily with the manufacturer's controls |
| (6) Hospital del Niño and Hospital Materno-Infantil, La Paz, Bolivia | Diazo method (Synchron CX5 Beckman Coulter, Inc, Fullerton, CA, USA) | Quality control was run once daily with a known sample provided by the manufacturer |

data are particularly lacking.[28] Infants presenting for care could only be enrolled once and, to ensure follow-up, had to reside in the defined study area. We excluded young infants who: (1) needed immediate cardiopulmonary resuscitation on presentation to the hospital, (2) had an obvious lethal congenital abnormality, (3) had been hospitalised in the previous 2 weeks (except for delivery) or (4) were referred from another health facility.

### Clinical assessment of jaundice

Enrolled subjects were referred to a trained PHW (ie, a front-line professional health worker, typically a nurse) for initial evaluation. Study coordinators worked with site principal investigators to standardise their clinical assessments within and across sites. Site principal investigators in turn trained, supervised and attempted to standardise PHWs in their assessments of jaundice. PHWs assessed infants unclothed under well-lit conditions for the presence of jaundice in three body regions: (1) head, including face, gums or sclera, (2) trunk, including chest or abdomen and (3) distal extremities, including palms or soles. If jaundice was present at any of the sites within a body region, the degree of jaundice was scored as mild or deep, taking the deepest level of staining for the sites within the body region. At Durban, the degree of jaundice was scored as mild, moderate or severe, but for analysis, moderate and severe degrees of jaundice were combined to form the 'deep' category. Sites did not start to enroll subjects at the same time; therefore, a change in study protocol part way through the study meant that in Kumasi - which was the first site to start and complete enrollment - limited data were available from the face and chest/abdomen in comparable form to that sought later at other study sites. Data from the La Paz study location were restricted to the first half of data collection, since the distribution of serum bilirubin levels in the second half of the study suggested the introduction of a systematic error, which could not be identified and addressed, despite quality control measures; thus, data of suspect quality were excluded.

After the initial PHW clinical assessment, the infant was referred to a study physician for immediate evaluation and management. The physician took a complete history and performed a physical examination, blinded to the PHW's findings and to laboratory data. Signs of jaundice were assessed by physicians using the same proforma as the PHWs.

### Laboratory evaluation of serum bilirubin

About 30–60 min after the study physician's evaluation, a blood sample was drawn from infants found to have any visible jaundice by the study physician and was tested for total serum bilirubin in the hospital laboratories (table 1).

### Patient management

Study physicians decided on the need for admission based on their clinical judgement and the serum bilirubin level. Admitted infants were managed according to the usual practices of the study hospitals.[29]

### Data management and analysis

Case record forms were checked for completion and double entered into an EpiData database (V.2.1, EpiData Association, Odense, Denmark) at each of the study locations. Data files were sent to the data coordination centre in Melbourne, Australia, where further data cleaning and consistency checks were performed, and the quality of data submitted from the individual study locations was monitored. All cases for which there was a physician assessment were included.

### Description of jaundice by physicians

Box-and-whisker plots were used to display the association between increasing depth and peripheral extension of jaundice (from face to trunk to distal extremities), as assessed by physicians, and serum bilirubin level. Four categories of staining were created for display: A=no jaundice, including the head; B=mild jaundice of the head or trunk, but none of the distal extremities; C=deep jaundice of the head or trunk, but none of the distal extremities (these data were not available for Kumasi) and D=Aay jaundice of the distal extremities.

### Sensitivity and specificity of jaundice definitions

A series of definitions was created based on the body site and depth of jaundice, including: (1) deep jaundice of the head, (2) deep jaundice of the trunk, (3) any jaundice of the distal extremities (palms, soles), (4) any jaundice of the distal extremities OR deep jaundice of the trunk and (5) any jaundice of the distal extremities OR deep jaundice of any site, including trunk or head. The sensitivity and specificity of classification according to these definitions of jaundice, by the PHWs and by the study physicians, were determined using thresholds for serum bilirubin as the gold-standard measure as follows: (1) >260 µmol/L (>15 mg/dL), which approximates the need for phototherapy according to WHO guidelines and (2) >340 µmol /L (>20 mg/dL), which approximates need for emergency intervention in at-risk and preterm babies.[28] All available data from PHW and physician assessments were used. Cases were classified as positive on each of the definitions assessed if any of the signs included in the definition were recorded; otherwise, they were classified as negative.

### Agreement between physicians and PHWs

Agreement between PHWs and physicians for identification of jaundice at each of the body regions and at each study location, for mild or severe degree of staining, was assessed by tabulating the frequency of positive classifications by each rater and using the kappa statistic. Patients aged 1–20 days were included regardless of whether they had any sign of jaundice on examination by physicians or a serum bilirubin measure. Selection of this age range was predetermined to address the gap in IMCI guidelines

on clinical signs of jaundice that signal urgent referral for hospital-level care in infants aged 1–20 days.

All data analyses were performed using Stata software, Release V.10 (StataCorp, College Station, Texas, USA).

### Data sharing
The study protocol, deidentified participant data and statistical code will be made available on publication of the manuscript.

### Funding
The United States Agency for International Development (USAID) provided funding for this study to the Applied Research on Child Health and Child and Family Applied Research projects at Boston University, Boston, by means of the USAID cooperative agreements (HRN-A-00-90010-00 and GHS-A-00-00020-00). The Bill and Melinda Gates Foundation provided funding for Saving Newborn Lives.

### Patient and public involvement
Families of study subjects were not involved in study design, recruitment or conduct or in the dissemination of study results.

### RESULTS
### Prevalence of jaundice and hyperbilirubimia
Among the 5250 young infants, 1–59 days of age who were enrolled in the parent trial at the six study sites,[1] 2642 (50%) were 1–20 days of age; 1018 (38% of 2642) were young neonates <1 week of age (days 1–6) and 1442 infants (62% of 2642) were 7–20 days of age. Dhaka had a majority (60%) of infants 1–6 days of age while Durban had a preponderance (85%) of infants 7–20 days of age (table 2). Among infants, 1–20 days of age, 8% (n=196) were preterm, ranging from 3% in La Paz to 11% in Delhi. Online supplemental table S1 provides additional information on the primary diagnoses of the 2642 infants 1–20 days of age.

Physicians assessed that 1337 of 5250 infants (25.5%) had any sign of jaundice, 1241 of whom were aged 1–20 days (47% of 2642 babies in that age range, 93% of all young infants with jaundice) (table 2). A total of 962 (78%) had a serum bilirubin measurement recorded, and among infants whose bilirubin level was measured, 278 (29%) and 95 (10%) had hyperbilirubinaemia >260 µmol /L (>15 mg/dL) and >340 µmol /L (>20 mg/dL), respectively. The prevalence of hyperbilirubinaemia varied substantially between study locations (>260 µmol/L: 10%–58%, >340 µmol/L: 2%–22%). Among 2608 infants aged 21–59 days (not included in this validation study), 96 (3.7%) had any sign of jaundice.

### Prevalence of jaundice by depth of staining
Distributions of depth of staining by body region, according to physicians, were generally similar across locations except that the Indian physicians (especially in Delhi) recorded more deep jaundice among those found to have any jaundice (table 3). In general, across study locations, there were decreasing proportions of subjects

**Table 2** Study profile: infants aged 1–20 days assessed for jaundice and, among eligible neonates, numbers with serum bilirubin measured and with hyperbilirubinaemia, by study location and overall

| | Study location | | | | | | Overall |
|---|---|---|---|---|---|---|---|
| | Dhaka | Delhi | Chandigarh | Durban | Kumasi | La Paz | |
| Number assessed for jaundice* | 448 | 330 | 700 | 414 | 457 | 293 | 2642 |
| Preterm birth | 42 | 36 | 36 | 42 | 30 | 10 | 196 |
| % of infants | 9% | 11% | 5% | 10% | 7% | 3% | 7% |
| Age group strata | | | | | | | |
| 1–6 days | 268 | 94 | 347 | 61 | 148 | 100 | 1018 |
| % of infants | 60% | 28% | 50% | 15% | 32% | 34% | 39% |
| 7–20 days | 180 | 236 | 353 | 353 | 309 | 193 | 1624 |
| % of infants | 40% | 72% | 50% | 85% | 68% | 66% | 61% |
| Any jaundice observed by study physicians | 198 | 148 | 398 | 160 | 147 | 190 | 1241 |
| % of assessed | 44% | 45% | 57% | 39% | 32% | 65% | 47% |
| Serum bilirubin obtained | 81 | 119 | 379 | 132 | 71 | 180 | 962 |
| % of those with jaundice | 41% | 80% | 95% | 91% | 48% | 95% | 78% |
| Serum bilirubin >260 µmol /L (15 mg/dL) | 36 | 62 | 39 | 19 | 18 | 104 | 278 |
| % of those with bilirubin measured | 44% | 52% | 10% | 14% | 25% | 58% | 29% |
| Serum bilirubin >340 µmol /L (20 mg/dL) | 14 | 26 | 9 | 2 | 8 | 36 | 95 |
| % of those with bilirubin measured | 17% | 22% | 2% | 2% | 11% | 20% | 10% |

*By both primary healthcare workers and physicians.

**Table 3** Clinical signs of jaundice by body region and depth of staining among neonates aged 1–20 days with any jaundice observed by physicians, by study location

| | Infants (%) with jaundice | | | | | |
| --- | --- | --- | --- | --- | --- | --- |
| | Study location | | | | | |
| Body region | Dhaka | Delhi | Chandigarh | Durban | Kumasi* | La Paz |
| Head: face, gums or sclera (n) | 81 | 119 | 379 | 132 | * | 180 |
| Mild | 64% | 19% | 46% | 78% | | 64% |
| Deep | 36% | 81% | 54% | 22% | | 36% |
| Trunk: chest or abdomen (n) | 81 | 119 | 379 | 115 | * | 180 |
| None | 26% | 8% | 24% | 17% | | 12% |
| Mild | 63% | 40% | 61% | 68% | | 80% |
| Deep | 11% | 51% | 15% | 15% | | 8% |
| Distal extremities: palms or soles (n) | 81 | 118 | 379 | 132 | 70 | 179 |
| None | 64% | 64% | 93% | 88% | 66% | 93% |
| Mild | 35% | 31% | 6% | 9% | 34% | 4% |
| Deep | 1% | 5% | 1% | 3% | 0% | 3% |

*At this site, assessments were not graded as "Mild" or "Deep"; for Palms/soles the "Mild" category includes all cases with any jaundice. A change in study protocol part way through the study meant that in Kumasi, which was the first site to start and complete enrollment, limited data were available from the face and chest/abdomen in comparable form to the other study sites

with deep jaundice when considering the head (22%–81%) compared with the trunk (8%–51%) and the distal extremities (palms/soles) (0%–5%).

### Association of clinical signs of jaundice with hyperbilirubinaemia

#### Descriptive analysis

At all study locations, higher serum bilirubin levels were measured as severity and distal distribution of jaundice increased (figure 1). However, this trend was more pronounced for some study locations (Delhi, Chandigarh) than for others (Kumasi), and the absolute levels

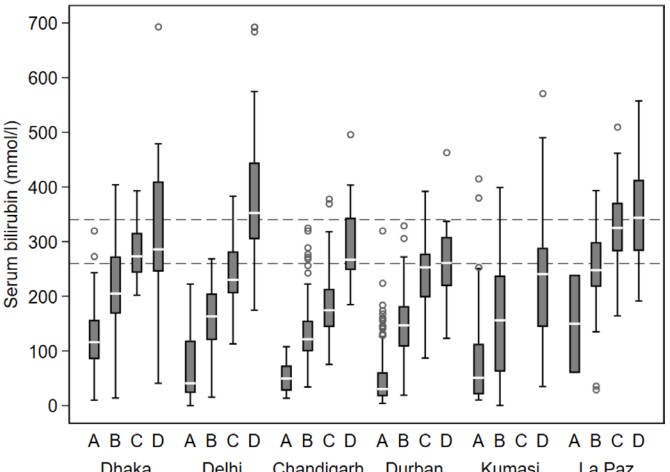

**Figure 1** Box plots of serum bilirubin levels in neonates aged 1–20 days according to clinical signs of jaundice detected by physicians, by study location. Horizontal lines indicate the two thresholds for hyperbilirubinaemia used in subsequent analysis (260 umol/L (15 mg/dL) and 340 (20 mg/dL) umol/L).

of serum bilirubin corresponding to the different clinical jaundice categories varied between locations. At Dhaka, Durban and La Paz, there was relatively little difference in serum bilirubin levels between patients with deep jaundice of the head or of the trunk, but none of the distal extremities (category C) compared with patients with jaundice extending to the distal extremities (category D).

#### Sensitivity and specificity

For the lower threshold of serum bilirubin (>260 μmol/L or >15 mg/dL), the highest overall sensitivity across sites was found for the definition 'any jaundice of the distal extremities (palms/soles) OR deep jaundice of the trunk or head' (PHWs: 58%–93% (table 4), physicians: 55%–98% (table 5)). Sensitivity for the algorithm 'any jaundice of the distal extremities' was lower for PHWs (42%–88%, table 4), and especially for physicians (11%–66%, table 5). Sensitivity was also lower for 'deep jaundice of the trunk' as reported by PHWs (26%–66%, table 4) and physicians (14%–81%, table 5). Sensitivity of 'deep jaundice of the head' was moderate for PHWs (32%–90%, table 4) and physicians (53%–97%, table 5). Sensitivity of physician assessments was particularly low across the various definitions for jaundice at La Paz (11%–55%, table 5).

Specificity of PHW assessments (table 4) was highest for 'deep jaundice of the trunk' (78%–96%) and ranged from 48%–89% across all other jaundice algorithms at all study locations. Specificity varied across sites for the algorithm 'any jaundice of the distal extremities or deep jaundice of the trunk or head' (46%–84%). Specificity of assessments by physicians (table 5) was relatively high across all algorithms at all sites (74%–100%) except at the Indian sites for 'deep jaundice of the head' and 'any

**Table 4** Sensitivity and specificity (as percent), with 95% CIs (in parentheses) of clinical signs of jaundice in neonates aged 1–20 days as assessed by primary healthcare workers for detecting levels of hyperbilirubinaemia

| | Sensitivity | | | | | | Specificity | | | | | |
|---|---|---|---|---|---|---|---|---|---|---|---|---|
| | Dhaka | Delhi | Chandigarh | Durban | Kumasi | La Paz | Dhaka | Delhi | Chandigarh | Durban | Kumasi | La Paz |
| **Serum bilirubin > 260 mmol/l (15 mg/dl)** | | | | | | | | | | | | |
| | (n=35)* | (n=61) | (n=38) | (n=18) | (n=17) | (n=103) | (n=41) | (n=54) | (n=294) | (n=103) | (n=46) | (n=78) |
| Deep jaundice of the head | 51 (34 to 69) | 90 (80 to 96) | 32 (18 to 49) | 56 (31 to 78) | | 75 (65 to 83) | 66 (49 to 80) | 48 (34 to 62) | 89 (85 to 92) | 80 (71 to 87) | | 49 (37 to 60) |
| Deep jaundice of the runk | 31 (17 to 49) | 66 (52 to 77) | 26 (13 to 43) | 47 (21 to 73) | | 31 (22 to 41) | 78 (62 to 89) | 81 (69 to 91) | 96 (93 to 98) | 82 (72 to 90) | | 83 (73 to 91) |
| Any jaundice of the distal extremities | 66 (48 to 81) | 64 (51 to 76) | 55 (38 to 71) | 56 (31 to 78) | 88 (64 to 99) | 42 (32 to 52) | 66 (49 to 80) | 80 (66 to 89) | 89 (85 to 92) | 56 (46 to 66) | 76 (61 to 87) | 71 (59 to 80) |
| Any jaundice of the distal extremities OR deep jaundice of the head | 66 (48 to 81) | 77 (65 to 87) | 58 (41 to 74) | 67 (41 to 87) | | 44 (34 to 54) | 66 (49 to 80) | 72 (58 to 84) | 88 (84 to 92) | 49 (39 to 59) | 69 (58 to 79) | 69 (58 to 79) |
| Any jaundice of the distal extremities OR deep jaundice of the head or trunk | 69 (51 to 83) | 93 (84 to 98) | 58 (41 to 74) | 83 (59 to 96) | | 80 (71 to 87) | 63 (47 to 78) | 48 (34 to 62) | 84 (79 to 88) | 48 (38 to 58) | | 46 (35 to 58) |
| **Serum bilirubin > 340 mmol/l (20 mg/dl)** | | | | | | | | | | | | |
| | (n=14)† | (n=25) | (n=9) | (n=2)‡ | (n=7) | (n=35) | (n=62) | (n=90) | (n=323) | (n=119) | (n=56) | (n=146) |
| Deep jaundice of the head | 79 (49 to 95) | 96 (80 to 100) | 67 (30 to 93) | | | 94 (81 to 99) | 66 (53 to 78) | 34 (25 to 45) | 88 (84 to 92) | 76 (67 to 83) | | 42 (34 to 51) |
| Deep jaundice of the trunk | 64 (35 to 87) | 76 (55 to 91) | 44 (14 to 79) | | | 46 (29 to 63) | 82 (70 to 91) | 66 (55 to 75) | 94 (91 to 97) | 77 (68 to 85) | | 80 (73 to 86) |
| Any jaundice of the distal extremities | 93 (66 to 100) | 80 (59 to 93) | 89 (52 to 100) | | 100 (59 to 100) | 57 (39 to 74) | 61 (48 to 73) | 67 (56 to 76) | 86 (81 to 89) | 55 (45 to 64) | 66 (52 to 78) | 68 (60 to 76) |
| Any jaundice of the distal extremities OR deep jaundice of the head or trunk | 93 (66 to 100) | 88 (69 to 97) | 89 (52 to 100) | | | 60 (42 to 76) | 61 (48 to 73) | 56 (45 to 66) | 85 (80 to 89) | 46 (37 to 56) | | 67 (59 to 75) |
| Any jaundice of the distal extremities OR deep jaundice of the head or trunk | 93 (66 to 100) | 100 (86 to 100) | 89 (52 to 100) | | | 97 (85 to 100) | 58 (45 to 70) | 33 (24 to …) | 81 (76 to 85) | 44 (35 to 53) | | 38 (30 to 47) |

*Numbers apply to serum bilirubin >260 mmol/L (15 mg/dL) for sites.
†Numbers apply to serum bilirubin >340 mmol/L (20 mg/dL) for sites.
‡Both cases were found to have deep jaundice of the face, and one also had jaundice of palms and soles.

**Table 5** Sensitivity and specificity (as percent), with 95% CIs. (in parentheses) of clinical signs of jaundice in neonates aged 1–20 days as assessed by physicians for detecting levels of hyperbilirubinaemia

| | Sensitivity | | | | | | Specificity | | | | | |
|---|---|---|---|---|---|---|---|---|---|---|---|---|
| | Dhaka | Delhi | Chandigarh | Durban | Kumasi | La Paz | Dhaka | Delhi | Chandigarh | Durban | Kumasi | La Paz |
| **Serum bilirubin > 260 mmol/l (15 mg/dl)** | | | | | | | | | | | | |
| | (n=36)* | (n=61) | (n=39) | (n=19) | (n=17) | (n=103) | (n=46) | (n=57) | (n=340) | (n=113) | (n=53) | (n=76) |
| Deep jaundice of the head | 53 (35 to 70) | 97 (89 to 100) | 85 (69 to 94) | 74 (54 to 94) | | 55 (45 to 65) | 78 (64 to 89) | 37 (24 to 51) | 50 (44 to 55) | 88 (80 to 93) | | 89 (80 to 95) |
| Deep jaundice of the trunk | 19 (8 to 36) | 81 (69 to 90) | 79 (64 to 91) | 44 (20 to 70) | | 14 (8 to 23) | 96 (85 to 99) | 81 (68 to 90) | 92 (89 to 95) | 90 (82 to 95) | | 100 (95 to 100) |
| Any jaundice of the distal extremities | 56 (38 to 72) | 66 (52 to 77) | 38 (23 to 55) | 42 (20 to 67) | 59 (33 to 82) | 11 (5 to 18) | 80 (66 to 91) | 95 (85 to 99) | 97 (94 to 98) | 93 (87 to 97) | 74 (60 to 85) | 99 (93 to 100) |
| Any jaundice of the distal extremities OR deep jaundice of the trunk | 56 (38 to 72) | 90 (80 to 96) | 79 (64 to 91) | 47 (24 to 71) | | 15 (9 to 24) | 80 (66 to 91) | 79 (66 to 89) | 91 (88 to 94) | 87 (79 to 92) | | 99 (93 to 100) |
| Any jaundice of the distal extremities OR deep jaundice of the head or trunk | 64 (46 to 79) | 98 (91 to 100) | 85 (69 to 94) | 79 (54 to 94) | | 55 (45 to 65) | 74 (59 to 86) | 37 (24 to 51) | 49 (44 to 55) | 83 (75 to 90) | | 88 (79 to 94) |
| **Serum bilirubin > 340 mmol/l (20 mg/dl)** | | | | | | | | | | | | |
| | (n=14)† | (n=26) | (n=9) | (n=2)‡ | (n=7) | (n=36) | (n=68) | (n=92) | (n=370) | (n=130) | (n=63) | (n=143) |
| Deep jaundice of the head | 79 (49 to 95) | 100 (87 to 100) | 100 (66 to 100) | | | 81 (64 to 92) | 74 (61 to 83) | 25 (16 to 35) | 47 (42 to 53) | 79 (71 to 86) | 75 (67 to 82) | 75 (67 to 82) |
| Deep jaundice of the trunk | 50 (23 to 77) | 92 (75 to 99) | 100 (66 to 100) | | | 28 (14 to 45) | 97 (90 to 100) | 60 (50 to 70) | 87 (83 to 90) | 85 (77 to 91) | | 97 (92 to 99) |
| Any jaundice of the distal extremities | 86 (57 to 98) | 85 (65 to 96) | 78 (40 to 97) | | 71 (29 to 96) | 17 (6 to 33) | 75 (63 to 85) | 77 (67 to 85) | 95 (92 to 97) | 88 (82 to 93) | 70 (57 to 81) | 96 (91 to 98) |
| Any jaundice of the distal extremities OR deep jaundice of the head or trunk | 86 (57 to 98) | 100 (87 to 100) | 100 (66 to 100) | | | 28 (14 to 45) | 75 (63 to 85) | 55 (44 to 65) | 86 (82 to 90) | 82 (75 to 88) | | 95 (90 to 98) |
| Any jaundice of the distal extremities OR deep jaundice of the head or trunk | 93 (66 to 100) | 100 (87 to 100) | 100 (66 to 100) | | | 81 (64 to 92) | 68 (55 to 78) | 24 (15 to 34) | 47 (42 to 52) | 75 (67 to 83) | | 74 (66 to 81) |

*Numbers apply to serum bilirubin >260mmol/L (15mg/dL) for sites.
†Numbers apply to serum bilirubin >340mmol/L (20mg/dL) for sites.
‡Both cases were found to have deep jaundice of the face, and one also had jaundice of palms and soles.

**Table 6** Agreement between primary healthcare workers and physicians in detecting clinical signs of jaundice in neonates aged 1–20 days, by study site

|  | Study location | | | | | |
|---|---|---|---|---|---|---|
|  | Dhaka | Delhi | Chandigarh | Durban | Kumasi | La Paz |
| Deep jaundice of the head | 488 | 334 | 703 | 414 |  | 293 |
|  | 0.14 0.12 | 0.27 0.31 | 0.06 0.3 | 0.09 0.07 |  | 0.43 0.24 |
|  | 0.77 | 0.82 | 0.23 | 0.5 |  | 0.42 |
| Deep jaundice of the trunk (chest or abdomen) | 488 | 334 | 703 | 323 |  | 293 |
|  | 0.08 0.02 | 0.15 0.19 | 0.03 0.08 | 0.08 0.05 |  | 0.16 0.05 |
|  | 0.43 | 0.73 | 0.35 | 0.61 |  | 0.44 |
| Any jaundice of the distal extremities (palms or soles) | 488 | 333 | 703 | 414 | 456 | 292 |
|  | 0.15 0.10 | 0.16 0.13 | 0.08 0.04 | 0.17 0.04 | 0.16 0.12 | 0.24 0.04 |
|  | 0.61 | 0.68 | 0.41 | 0.29 | 0.49 | 0.21 |

jaundice of the distal extremities OR deep jaundice of the trunk or head' (37%–50%).

For the higher threshold of hyperbilirubinaemia (>340 μmol/L or >20 mg/dL), sensitivity was higher and specificity was similar to the findings at the lower serum bilirubin threshold for PHWs (table 4) and physicians (table 5). High sensitivity was found for 'any jaundice of the distal extremities OR deep jaundice of the trunk or head' for PHWs (89%–100%, table 4) and physicians (81%–100%, table 5). Specificity for this algorithm was more variable for both PHWs (33%–81%, table 4) and physicians (24%–75%, table 5). A reasonable balance of sensitivity (71%–100%) and specificity (55%–95%) across both PHW and physician assessments was observed, excluding La Paz, for the definition 'any jaundice of distal extremities.'

### Agreement between PHWs and physicians in assessment of jaundice

Agreement between PHWs and physicians was variable for specific definitions of jaundice and across sites (table 6). Agreement between PHWs and physicians in identifying neonates with any signs of jaundice on the head was generally good but agreement for depth of jaundice and for jaundice of the extremities was variable. Overall agreement was highest for Delhi (0.68–0.82 across jaundice algorithms).

Data for each definition of clinical jaundice are displayed as follows: (row 1) number of infants evaluated by both observers; (row 2) proportion of infants judged positive by PHWs and by physicians, respectively; (row 3) kappa statistic for measuring agreement beyond chance.

### DISCUSSION

The study is the largest evaluation reported to date on the validity of clinical assessments for jaundice by PHWs and physicians in identifying neonates with hyperbilirubinaemia at levels indicating need for referral-level care. We observed that a finding of 'any jaundice of the distal extremities' (palms or soles) by PHWs or physicians had

sensitivity and specificity generally exceeding about 70% across locations for detecting neonates aged 1–20 days with bilirubin levels >340 μmol /L (20 mg/dL)—a level that requires acute intervention in a health facility, including evaluation for plasma exchange transfusion in at-risk and preterm babies. Given that this level of jaundice places infants at risk of severe morbidity, we also note that the broader algorithm 'any jaundice of the distal extremities OR deep jaundice of the trunk or head' had high sensitivity for both PHWs (89%–100%) and physicians (81%–100%), although with variable specificity across study sites (33%–81% for PHWs and 24%–75% for physicians). Use of this latter algorithm could result in the referral of not only more at-risk infants (based on high sensitivity), but also, depending on specificity at the site, more infants without significant hyperbilirubinaemia. This latter algorithm may be preferable in settings, in which prompt referral and access to timely measurement and management of hyperbilirubinaemia are lacking or anticipated to be delayed, such as in many LMIC peripheral healthcare settings with minimal equipment and skilled providers where use of the higher sensitivity algorithm may minimise missing patients who can benefit from initiation of treatment while referral is managed. Clinical signs were also discriminatory, but less so, for identifying infants who had moderately elevated bilirubin (>260 μmol/L). For this lower threshold level indicating need for phototherapy, the algorithm 'any jaundice of the distal extremities OR deep jaundice of the trunk or head' also had the highest overall sensitivity across sites (58%–93% for PHWs and 55%–98% for physicians). Thus, in settings where it is not possible to measure serum bilirubin, such as in many peripheral health facilities in LMICs, which use IMCI, it is recommended to refer neonates aged 1–20 days with yellow palms and soles or deep jaundice on the trunk or head to a hospital for further evaluation and management.

We found very few cases in which jaundice appeared on the first day after delivery or lasted beyond 3 weeks of age; all these babies should be referred to a health facility

for further assessment and management, including direct and total bilirubin measurement. A small percentage of infants aged 21–59 days had jaundice (n=96, 3.7%); jaundice presenting in this age group is presumed pathological and all such infants should also be referred for further clinical and laboratory workup.

Several previous studies have assessed the value of clinical signs for predicting high levels of serum bilirubin in neonates.[10–19] In many LMICs where births commonly occur either at home or at peripheral health clinics, laboratory measurement of serum bilirubin may not be easily available and clinical signs have to be used as a basis for referral. Similar to some previous studies in LMICs, we found that serum bilirubin levels were generally higher in neonates with jaundice of more caudal parts of the body.[12–16] Jaundice extending most distally to the palms and soles was most discriminatory in identifying neonates with very high levels of bilirubin in need of urgent evaluation for treatment, and thus, has clinical relevance. However, jaundice that deeply involved the trunk or head was also useful clinically for identification of patients with hyperbilirubinaemia requiring treatment and recognition of this more sensitive sign, as part of the algorithm 'any jaundice of the distal extremities or deep jaundice of the trunk or head' may also be important in low resource settings with fewer diagnostic and treatment capabilities.

In general, PHWs had higher sensitivity than physicians in identifying neonates with hyperbilirubinaemia based on the identification of clinical signs of jaundice. Overall, the sensitivity of any jaundice on palms and soles in detecting infants with high bilirubin levels >340 µmol/L was higher for PHWs than for physicians, but the specificity was about 10 percentage points lower in most locations. For identifying infants with bilirubin levels >260 µmol/L, specificity was slightly higher for PHWs, while sensitivity was similar for both cadres. These results emphasise that once neonates with suspected hyperbilirubinaemia based on PHW clinical assessment reach a health facility, further physician clinical assessment and laboratory evaluations must be performed whenever possible to determine need for treatment. We observed considerable variation in the performance of clinical signs of jaundice between study locations, despite attempting to standardise PHW training and physician supervision. Overall, clinical signs had lower sensitivity in La Paz and lower specificity in Dhaka and Durban, and agreement for the presence of jaundice on the palms and soles was lowest in Durban and La Paz compared with the other locations. These findings highlight the importance of local evaluation and adaptations of algorithms.

Variable skin pigmentation is an important potential confounding factor in the clinical detection of jaundice.[13] In Kumasi, where skin pigmentation was darkest among the study locations, study investigators were not confident that they could identify jaundice on the face, chest or thighs; as a result, their assessments of jaundice were limited to the sclera, gums, palms and soles. Nevertheless, PHWs in Kumasi were able to identify neonates needing referral for hyperbilirubinaemia based on recognition of jaundice on the palms and soles, with good sensitivity and reasonable specificity.

This study had several strengths including its size (n=1242) and multiple sites, jaundice assessment at several locations on each infant and systematic evaluation of PHW assessments of jaundice. The findings have the potential to be generalised to large populations in LMICs.

This study had several limitations. We did not attempt to objectively assess the depth of jaundice, for example, by the use of colour scales for comparison. This reflects the reality in many LMIC settings, where such scales or visual cue cards are not available; however, their use could be a valuable addition in future research. Also, due to inconsistencies in obtaining blood samples for bilirubin measurement, 22% of infants with some degree of jaundice did not have laboratory work done. In addition, the decision to draw blood for bilirubin detection was based on the study physicians' diagnoses and not the PHWs' assessments. Given that PHWs had higher sensitivity than physicians in identifying neonates with hyperbilirubinaemia based on the identification of clinical signs of jaundice, some infants with jaundice identified by PHWs did not have blood samples obtained. Since the study design focused on sick young infants presenting for care, asymptomatic infants with jaundice may not have been brought for assessment, and, thus, the sensitivity and specificity of clinical assessments may vary at population level and warrant further investigation. However, infants presenting for routine healthcare (eg, well baby checks) may also be at risk for complications of jaundice and, therefore, should be screened. We did not identify the characteristics of the newborns who are prone to screening errors, which we recommend for future research to further improve algorithm performance. This study did not include sufficient numbers of preterm infants to allow for independent assessment of the utility of jaundice detection in this subgroup of infants, who are especially vulnerable to morbidity and mortality from hyperbilirubinaemia. It is challenging to accurately identify gestational age in LMICs where ultrasound assessment may not be available. Therefore, a more conservative approach to referral of jaundiced infants suspected to be preterm is advisable, for example, use of the algorithm 'any jaundice of the distal extremities or deep jaundice of the trunk or head'. Finally, there were wide variations between sites in the proportion of jaundiced infants who had a bilirubin measured (41%–95%) and in the proportion of bilirubin levels that were severe (2%–22% >340 µmol/L). Reasons for variation in severity of hyperbilirubinaemia by site are not known but could reflect local differences in underlying risk factors as well as access to and timeliness of care seeking. While the potential impact of these issues on our findings is unclear, current estimates of sensitivity and specificity of visual appraisal for identifying infants with various thresholds of hyperbilirubinaemia should be interpreted with some caution.

## CONCLUSION

Further standardisation of training to ensure greater consistency among PHWs' assessments as well as local research and monitoring in settings, where IMCI guidelines are implemented, would be useful to further refine the assessment and referral criteria for jaundice. The current IMCI algorithm focuses on jaundice of palms and soles as indicative of need for urgent referral. However, the more broadly defined algorithm 'any jaundice of the distal extremities deep jaundice of the trunk or head' may identify more at-risk neonates—although at a potential cost of referral of greater numbers of infants not needing acute intervention (eg, phototherapy, exchange transfusion)—and may be preferred in settings where delay in receiving care is anticipated. Local evidence-based adaptations of these algorithms are needed, particularly regarding consideration of deep staining of the trunk or head and the balance between increased sensitivity, enabling the detection of greater numbers of infants needing further assessment and increased specificity and targeting of scare diagnostic and treatment resources. Many hospitals in LMICs cannot determine bilirubin levels reliably.[30] In addition to building basic laboratory capacity in developing country hospitals, further refinement and evaluation of the utility of point-of-care technologies[20–22 30] to augment clinical assessment for detection of hyperbilirubinaemia is warranted.

**Author affiliations**
[1]Department of Pediatrics, Stanford University School of Medicine, Stanford, California, USA
[2]Department of Global Health, Boston University School of Public Health, Boston, Massachusetts, USA
[3]Section of Infectious Diseases, Department of Medicine, Boston University School of Medicine, Boston, MA, USA
[4]Clinical Epidemiology and Biostatistics Unit, Murdoch Children's Research Institute & The University of Melbourne, Melbourne, Victoria, Australia
[5]Department of Paediatrics and Child Health, University of KwaZulu-Natal Nelson R Mandela School of Medicine, Durban, South Africa
[6]Department of Pediatrics, Hospital del Nino Dr Ovidio Aliaga Uria, La Paz, Plurinational State of Bolivia
[7]Departments of Pediatrics and Medical Microbiology, Post Graduate Institute of Medical Education and Research, Chandigarh, Chandigarh, India
[8]Department of Pediatrics, Division of Neonatology, All India Institute of Medical Sciences, New Delhi, Delhi, India
[9]School of Medical Sciences, Kwame Nkrumah University of Science and Technology, Kumasi, Ghana
[10]Department of Neonatology, Dhaka Shishu Hospital, Dhaka, Bangladesh
[11]Kwame Nkrumah University of Science and Technology, Kumasi, Ghana
[12]Department of Pediatrics, Hospital Materno-Infantil, La Paz, Plurinational State of Bolivia
[13]Child Health Research Foundation, Dhaka, Bangladesh
[14]Newborn Health Unit, Department of Maternal, Newborn, Child and Adolescent Health and Ageing, World Health Organization, Geneva, Switzerland
[15]World Health Organization Regional Office for Europe, Copenhagen, Denmark

**Acknowledgements** We thank the local communities at the study sites for their participation in the study, and also thank all members of the Young Infant Clinical Signs Study Group (YICSSG) who made this study possible. YICSSG members include (exclusive of named co-authors): Study sites: Bangladesh: Clinical investigators: A. S. M. Nawshad Uddin Ahmed, Md. Monir Hossain; Study physicians: Nazmun Nahar; Nurses: Amala Baidya, Mahmuda Parul; Laboratory personnel: Maksuda Islam, Tania Nasreen; Data management: Md. Rezaur Rahaman; Bolivia: Study physicians: Teresa Villagomez, Pablo Mattos; Manuel Pantoja Ludueña, Remedios Zumarán; Study nurses: Irma Quispe, Willy Tarqui, Lourdes Checa, Claudia Canqui; Data management: Erick Dueñas, Omar Vargas; Ghana: Study physicians:G. Plange-Rhule, Osei Akoto; Laboratory: M. Lartey; Data management: Henrietta Akpene; India, Chandigarh: Clinical investigators: Rupinder Narang; Study physicians: Prasad Muley, Satish Misra; Nurses: Tapasaya, Sanjay Rani; Laboratory: Pallab Ray, Tamanna Gaur; Data management: Vishal Kanojia, Ajay Dogra; India, Delhi: Clinical investigators: Harish Chellani, M. S. Prasad; Study physician: A. Satyavani; Nurses: Jyoti, Raji John; Laboratory: Arti Kapil; Data management: Sanjeev Negi, Narinder Singhal; South Africa: Clinical investigator: Mirian Adhikari; Nurse: Sister Mojaphelo; Laboratory: Wim Sturm; Data management: Precious Sikhakhane. Study advisors: Kim Mulholland, Vinod Paul, Eric Simoes, Jelka Zupan Data Management: Philip Greenwood, Claudine Chionh, Murdoch Children's Research Institute.

**Contributors** GLD, DH, JC and MW co-conceived the study design, planned and prepared study protocols, and provided oversight to overall implementation, data collection, data analysis, and interpretation of data. Site PIs provided input to study design, and directly coordinated and provided oversight to study implementation, data collection, data reporting, and interpretation of data at Chandigarh (PK, AN, PR), Delhi (AKD), Kumasi (EA-Y, YA-S, KY-A), Durban (PMJ), Bangladesh (MAKAC, SKS), and La Paz (AB and EM). RB contributed to data analysis and interpretation of data. JC led analysis of data across the study sites. EF contributed to literature review and interpretation of data. GLD, DH, JC and MW had access to raw data. GLD wrote the first draft of the manuscript and was responsible for the overall content as the guarantor. GLD accepts full responsibility for the work and/or the conduct of the study, had access to the data, and controlled the decision to publish. All authors reviewed and provided input to the manuscript and approved the paper for publication.

**Competing interests** RB and MW are WHO staff members. MW contributed to study design, implementation, data analysis and manuscript writing and RB contributed to data analysis and manuscript writing. The opinions expressed in this paper do not necessarily represent the position of the WHO. The funding agencies did not influence the conduct or outcomes of the study, data analysis or interpretation, or preparation of this paper. The authors declare no other potential competing interests.

**Patient consent for publication** Not applicable.

**Ethics approval** This study involves human participants and was approved by each of the sites' institutional review boards (IRB) or ethical committees. The Kumasi (number H-23160) and Durban (number H-22434) protocols were also approved by the Boston University Medical Center IRB. The Dhaka and La Paz protocols were reviewed by the Johns Hopkins University IRB and determined to be exempt from review because the protocol involved minimal risk to participants. The Delhi and Chandigarh protocols were approved by the WHO Secretariat Committee on Research Involving Human Subjects (number RPC 039/040). Informed consent was obtained for all participants. Participants gave informed consent to participate in the study before taking part.

**Provenance and peer review** Not commissioned; externally peer reviewed.

**Data availability statement** Data are available upon reasonable request. De-identified participant data and statistical code will be made available upon reasonable request.

**ORCID iDs**
Gary L Darmstadt http://orcid.org/0000-0002-7522-5824

Davidson H Hamer http://orcid.org/0000-0002-4700-1495
Praveen Kumar http://orcid.org/0000-0003-4742-8787
Kojo Yeboah-Antwi http://orcid.org/0000-0002-3516-9266
Rajiv Bahl http://orcid.org/0000-0001-7936-6985

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
