## [Reviewer comments · BMJ Open]

ARTICLE DETAILS

TITLE (PROVISIONAL)	Validation of visual estimation of neonatal jaundice in low-and middle-income countries: a multicentre observational cohort study
AUTHORS	Darmstadt, Gary; Hamer, D.; Carlin, John; Jeena, Prakash; Mazzi, Eduardo; Narang, Anil; Deorari, A; Addo-Yobo, Emmanuel; Chowdhury, MAK Azad; Kumar, Praveen; Abu-Sarkodie, Yaw; Yeboah-Antwi, Kojo; Ray, Pallab; Bartos, Andres; Saha, Samir K.; Foote, Eric; Bahl, Rajiv; Weber, Martin

VERSION 1 – REVIEW

REVIEWER	Kuniyoshi, Yasutaka Kensei Hospital, Department of Pediatrics
REVIEW RETURNED	10-Mar-2021

GENERAL COMMENTS	Thank you for the opportunity to review this paper. To determine neonatal jaundice, the authors have validated the use of clinical judgment along with visual assessment by primary healthcare workers and physicians in low- and middle-income countries. The study has a few limitations. Because the target population was limited only to neonates with acute illness, it is uncertain whether these results can be applied to all neonates, as pointed out by the authors. This research is not entirely novel because several studies on the same topic have already been carried out, as cited by the authors. However, this study contains a larger sample size than the previous studies. Major comments: 1. I could not find information about the participants' profiles. The authors have pointed out that the participants' reported gestational age is not sufficiently reliable (P22L7). However, if possible, the authors should also include data about the participants' birthweight, number of days after birth and weight at the time of assessment, and type of acute illness, because the presence of acute illness or body physique may affect the judgment of skin color.2. The authors have indicated the following: "A case was included if there was a response on at least one of the clinical signs; the signs that were not recorded were assumed absent.(P10L13)" Does this process lead to selection bias?
---

	Have both primary healthcare workers and physicians checked for the presence of jaundice in the following three body regions for all participants: 1) head, including face, gums, or sclera; 2) trunk, including chest or abdomen; and 3) distal extremities, including palms or soles? Minor comments:  1. If any characteristics in the neonates differed between the evaluators' assessment and serum bilirubin levels, the authors must provide this information. Identifying the characteristics of the newborns that are prone to screening errors will be helpful for the clinical staff. 2. The row names in Rows 12 and 13 are the same in Tables 4A and 4B. Please verify that these names are accurate. 3. It would be more appropriate to unify the expression "any jaundice of the distal extremities or deep jaundice of the trunk or head" in the text and "Any jaundice of distal extremities OR Deep jaundice of head or trunk" in Tables 4A and 4B. 4. Clarify whether "(n=XXX)" in Tables 4A and 4B is limited to Row 3 or Row 9.
--	--

REVIEWER	Liley, Helen The University of Queensland, Mater Research Institute
REVIEW RETURNED	30-Apr-2021

GENERAL COMMENTS	Thanks for the opportunity to review this interesting paper, which addresses a clinically important global health problem using a pragmatic approach. The study is a prospective, multicentre observational study in infants born in low or middle income countries of visual estimation of severity of jaundice when compared to serum bilirubin levels. The strengths of the study include the prospective, multicentre study design, central study coordination by an established consortium of site and central collaborators who have highly relevant expertise. Limitations are mostly well described in the final paragraph of the discussion. They include: Reliance on subjective judgements by the observers, without aids such as visual cue cards (although this may also be considered a strength, in that these circumstances may reflect the very low resource settings where the methods described in the study are most applicable). Exclusion of over a fifth of potentially eligible infants because no serum bilirubin determination was done, or because only the doctor's estimation was used to determine the need for serum bilirubin determination (noting that the study suggests that the
---

	doctors' estimations were not much better than the other healthcare workers). Potential selection bias towards sick infants and related to parental selection of which infants were brought to a participating hospital "for any type of acute illness". Possible selection bias towards term infants, as well as uncertainties about accuracy of gestational age determination. There was wide variation in the proportion of jaundiced infants who had a bilirubin measured between sites (more emphasis could be placed on this as a limitation). It is hard to estimate the overall potential of these issues to introduce bias or confounding. It seems likely that they could affect the estimates of sensitivity, specificity of the visual appraisal method at various thresholds, but may not negate the overall conclusions. Another issue is that there is no mention of whether the rates of severe jaundice in the different sites were expected to be similar, and whether the distribution of underlying etiologies was estimated to be similar or different. Specific comments on sections of the paper are as follows: Title (and other places): The term "clinical assessment" would be better replaced by "visual estimation". Abstract: Suggest minor amendment to "Infants referred or needing immediate cardiopulmonary resuscitation..." for clarity. Is this meant to be infants referred by another health service or practitioner for any reason or infants referred because they needed CPR? Why is plasma exchange specified? Most low resource settings would not have facilities for neonatal plasmapheresis. (Where facilities are available and in some underlying diseases, plasmapheresis may be an acceptable treatment, typically where severe jaundice is not accompanied by significant hemolysis). However, most exchange transfusions for hyperbilirubinemia are done using red cells or a combination of fractionated blood products (e.g., red cells plus plasma, depending on the hematocrit and suspension solution used for red cell preparation) because plasma exchange to the extent needed to remove clinically significant amounts of bilirubin in severe hyperbilirubinemia would risk causing or exacerbating anemia. It would be helpful for readers if non-standard abbreviations used in the abstract (e.g. PHWs) were spelled out. Section on Strengths and Limitations of this study: Bullet point 4 could state more explicitly that due to the characteristics of the larger study within which this one was embedded, there was likely participant selection bias towards infants who were considered unwell. Because sensitivity and specificity are affected by the rate of the outcome of interest in the population studied, the population level receiver operator characteristics will differ (i.e. change the word 'may'). (As mentioned above) the variation in study results between sites deserves mention. Introduction:
--	---

	Page 5, end of first paragraph – the risk of stillbirth relates to fetal anaemia and hydrops, not jaundice. Suggest change in wording to clarify this. Page 7, first paragraph – Low and middle income counties are not defined anywhere in the paper. Suggest using a citation for criteria. Same paragraph – suggest replacing “Point-of-care technologies” with “Point-of-care devices”. Methods: Page 6 1st paragraph – (minor point) presumably a parent provided informed consent, not the infants? The comment about “including sample size determination” is a bit vague. It is not clear to me that the 2008 provides a sample size estimation for the current study. More likely, the sample size for the current study was a sample of convenience, with the numbers available determined by the fact that it was nested in the 2003-2005 study. Is this correct? Table 1 the first row switches from the SI units used elsewhere in the paper to mg/dL. Also, the calibration range for this device (3.3-5.6 mg/dL) needs some further comment by the authors, because it is nowhere near the range of dangerous bilirubin levels. Do the authors have any confidence in its accuracy in the range of bilirubin levels above 20 mg/dL (~340 micromol/L). Is there confidence in the calibration ranges of the analysers used in other locations? “Calorimetric” in row 3 of the table should probably be “colorimetric” Clinical assessment of jaundice page 8 – “principle” should be “principal” The change in protocol and partial use of data from two sites deserves some additional explanation, perhaps in a supplementary document. Results: Prevalence of jaundice and hyperbilirubinemia: Given that the strata for the study were 1-6 and 7-50 days, why are results presented for 1-20 days? Was this an a priori decision? If it was post-hoc, is there risk of selective outcome reporting? In the same section, it should be clarified that the denominator for the percentages with high bilirubins is those babies who had a bilirubin measured, not all jaundiced infants. Page 17 2nd paragraph; the sentence “For the higher threshold of hyperbilirubinemia... sensitivity was higher and specificity was comparable to slightly lower compared to the findings at the lower serum bilirubin threshold...” would be clearer if the term “comparable to slightly lower” was replaced. Would “similar” be better? Discussion: Page 19 1st paragraph; The sentence: “This latter algorithm may be preferable in settings in which prompt referral and access to timely measurement and management of hyperbilirubinemia are lacking or likely to be delayed” may need improvement. Some explanation of “prompt referral” is needed because the study was done in (and is presumably applicable to) the hospitals where treatment would usually be provided. Is this not the case? Also, if access to measurement and management are not available at all, the threshold used is moot.
--	--

	The term “first level health facilities” should perhaps be better defined. On page 20, the term “peripheral health clinics” is used. Are these the same thing? Page 20 first paragraph; Re; the sentence “However jaundice that deeply involved the trunk or head was also useful clinically (reviewer question – for what?) and recognition of this less distal but more sensitive sign....may be particularly important...”. It seems self-evident that the trunk and head are “less distal” than palms or soles. However, the reason why this makes an algorithm using facial or truncal jaundice “particularly important” is very unclear. Is “algorithm” (a process or set of instructions to be followed) the best word? The term “algorithm” throughout the paper seems to be used to mean “detection site(s)”. References: In references 7, 8, 27 and 28, there is variation in how World Health Organization is abbreviated and some correction is needed.
--	--

VERSION 1 – AUTHOR RESPONSE

Reviewer: 1

Dr. Yasutaka Kuniyoshi, Kensei Hospital

Comments to the Author:

Thank you for the opportunity to review this paper.

To determine neonatal jaundice, the authors have validated the use of clinical judgment along with visual assessment by primary healthcare workers and physicians in low- and middle-income countries.

The study has a few limitations. Because the target population was limited only to neonates with acute illness, it is uncertain whether these results can be applied to all neonates, as pointed out by the authors. This research is not entirely novel because several studies on the same topic have already been carried out, as cited by the authors. However, this study contains a larger sample size than the previous studies.

Major comments:

1. I could not find information about the participants’ profiles. The authors have pointed out that the participants’ reported gestational age is not sufficiently reliable (P22L7). However, if possible, the authors should also include data about the participants’ birthweight, number of days after birth and weight at the time of assessment, and type of acute illness, because the presence of acute illness or body physique may affect the judgment of skin color.

Response: Information on preterm birth rates and age (by strata) has been added to Table 2, and text has been added to the Results as follows (added text in italics): “Among the 5250 young infants 1-59 days of age who were enrolled in the parent trial at the six study sites,¹ 2642 (50%) were 1-20 days of age; 1018 (38% of 2642) were young neonates < 1 week of age (days 1-6) and 1442 infants (62% of 2642) were 7-20 days of age. Dhaka had a majority (60%) of infants 1-6 days of age while Durban a had a preponderance (85%) of infants 7-20 days of age (Table 2). Among infants 1-20 days of age, 8% (n=196) were preterm, ranging from 3% in La Paz to 11% in Delhi. Table S1 provides additional information on the primary diagnoses of the 2642 infants 1-20 days of age.” In addition, a table of primary diagnoses of study participants has been added as Table S1.

2. The authors have indicated the following: “A case was included if there was a response on at least one of the clinical signs; the signs that were not recorded were assumed absent.(P10L13)” Does this process lead to selection bias?

Response: We have clarified the wording as it may have been confusing. This sentence now reads (added text in italics): “Cases were classified as positive on each of the definitions assessed if any of the signs included in the definition were recorded; otherwise they were classified as negative.”

Have both primary healthcare workers and physicians checked for the presence of jaundice in the following three body regions for all participants: 1) head, including face, gums, or sclera; 2) trunk, including chest or abdomen; and 3) distal extremities, including palms or soles?

Response: Yes, both primary healthcare workers and physicians checked for jaundice at these sites. This is noted in the last sentence of the section on Clinical assessment of jaundice, as follows: “Signs of jaundice were assessed by physicians using the same proforma as the PHWs.”

Minor comments:

1. If any characteristics in the neonates differed between the evaluators’ assessment and serum bilirubin levels, the authors must provide this information. Identifying the characteristics of the newborns that are prone to screening errors will be helpful for the clinical staff.

Response: Thank you for the interesting idea. The aim of our study was to document the quality of assessment, not to attempt to investigate possible reasons for misdiagnosis. While we are not able to address this, we have noted this for attention in future research in the Discussion, as follows (added text in italics): “We did not identify the characteristics of the newborns that are prone to screening errors, which we recommend for future research to further improve algorithm performance.”

2. The row names in Rows 12 and 13 are the same in Tables 4A and 4B. Please verify that these names are accurate.

Response: Primary healthcare workers and physicians assessed infants for the same set of signs of jaundice, so the row names for the signs are the same in Tables 4A (for primary healthcare workers) and 4B (for physicians).

3. It would be more appropriate to unify the expression “any jaundice of the distal extremities or deep jaundice of the trunk or head” in the text and “Any jaundice of distal extremities OR Deep jaundice of head or trunk” in Tables 4A and 4B.

Response: The language has been harmonized in the tables and text.

4. Clarify whether “(n=XXX)” in Tables 4A and 4B is limited to Row 3 or Row 9.

Response: The first set of numbers applies to sites for serum bilirubin > 260 mmol/l (15 mg/dl), and the second set of numbers applies to sites for serum bilirubin > 340 mmol/l (20 mg/dl). This has been clarified in the tables.

Reviewer 2

Prof. Helen Liley, The University of Queensland

Competing interests of Reviewer: No competing interests of significance

Thanks for the opportunity to review this interesting paper, which addresses a clinically important global health problem using a pragmatic approach. The study is a prospective, multicentre observational study in infants born in low or middle income countries of visual estimation of severity of jaundice when compared to serum bilirubin levels.

The strengths of the study include the prospective, multicentre study design, central study coordination by an established consortium of site and central collaborators who have highly relevant expertise.

Limitations are mostly well described in the final paragraph of the discussion. They include:

- Reliance on subjective judgements by the observers, without aids such as visual cue cards (although this may also be considered a strength, in that these circumstances may reflect the very low resource settings where the methods described in the study are most applicable).

Response: Mention of lack of visual cue card in low resource settings has been added to the last paragraph of the Discussion as follows (see italics): “We did not attempt to objectively assess the depth of jaundice, for example by the use of color scales for comparison. This reflects the reality in many LMIC settings where such scales or visual cue cards are not available, however, their use could be a valuable addition in future research to improve diagnostic algorithms for jaundice.”

- Exclusion of over a fifth of potentially eligible infants because no serum bilirubin determination was done, or because only the doctor’s estimation was used to determine the need for serum bilirubin determination (noting that the study suggests that the doctors’ estimations were not much better than the other healthcare workers).
- Potential selection bias towards sick infants and related to parental selection of which infants were brought to a participating hospital “for any type of acute illness”.
- Possible selection bias towards term infants, as well as uncertainties about accuracy of gestational age determination.
- There was wide variation in the proportion of jaundiced infants who had a bilirubin measured between sites (more emphasis could be placed on this as a limitation).

Response: The following point has been added to the last paragraph of the Discussion on limitations: “Finally, there were wide variations between sites in the proportion of jaundiced infants who had a bilirubin measured (41-95%) and in the proportion of bilirubin levels that were severe (2-22% >340 $\mu\text{mol/l}$). Reasons for variation in severity of hyperbilirubinemia by site are not known, but could reflect local differences in underlying risk factors as well as access to and timeliness of care-seeking.”

It is hard to estimate the overall potential of these issues to introduce bias or confounding. It seems likely that they could affect the estimates of sensitivity, specificity of the visual appraisal method at various thresholds, but may not negate the overall conclusions.

Response: As confounding is a particular source of bias only relevant to the estimation of causal effects, this is not an issue here. The threat of bias may be outweighed by the overall finding of the limitation of visual assessments. In this regard, the most pertinent bias would be one that would have artificially reduced the quality of the visual assessment, which we could not identify. Nevertheless, the following text has been added in acknowledgement of this possibility: “While the potential impact of these issues on our findings are unclear, current estimates of sensitivity and specificity of visual appraisal for identifying infants with various thresholds of hyperbilirubinemia should be interpreted with some caution.”

Another issue is that there is no mention of whether the rates of severe jaundice in the different sites were expected to be similar, and whether the distribution of underlying etiologies was estimated to be similar or different.

Response: This point has been addressed in the limitations section of the Discussion as follows: “Finally, there were wide variations between sites in the proportion of jaundiced infants who had a bilirubin measured (41-95%) and in the proportion of bilirubin levels that were severe (2-22% >340 $\mu\text{mol/l}$). Reasons for variation in severity of hyperbilirubinemia by site are not known, but could reflect local differences in underlying risk factors as well as access to and timeliness of care-seeking.”

Specific comments on sections of the paper are as follows:

1. Title (and other places): The term “clinical assessment” would be better replaced by “visual estimation”.

Response: Done

2. Abstract:

- a. Suggest minor amendment to “Infants referred or needing immediate cardiopulmonary resuscitation...” for clarity. Is this meant to be infants referred by another health service or practitioner for any reason or infants referred because they needed CPR?

Response: This has been clarified as follows (see italics): “Infants referred for any reason from another health facility or those needing immediate cardiopulmonary resuscitation were excluded.”

- b. Why is plasma exchange specified? Most low resource settings would not have facilities for neonatal plasmapheresis. (Where facilities are available and in some underlying diseases, plasmapheresis may be an acceptable treatment, typically where severe jaundice is not accompanied by significant hemolysis). However, most exchange transfusions for hyperbilirubinemia are done using red cells or a combination of fractionated blood products (e.g., red cells plus plasma, depending on the hematocrit and suspension solution used for red cell preparation) because plasma exchange to the extent needed to remove clinically significant amounts of bilirubin in severe hyperbilirubinemia would risk causing or exacerbating anemia.

Response: Thank you for this insightful comment. This has been modified to: “...need for emergency intervention ...”

- c. It would be helpful for readers if non-standard abbreviations used in the abstract (e.g. PHWs) were spelled out.

Response: The addition of words by spelling out primary healthcare workers would mean that valuable explanatory text would have to be deleted. We do not think this is a good trade-off. If the editor feels that it is important to spell out PHWs, we will go back and make other deletions in exchange to keep the word to <300. Thank you for your consideration.

3. Section on Strengths and Limitations of this study: Bullet point 4 could state more explicitly that due to the characteristics of the larger study within which this one was embedded, there was likely participant selection bias towards infants who were considered unwell. Because sensitivity and specificity are affected by the rate of the outcome of interest in the population studied, the population level receiver operator characteristics will differ (i.e. change the word ‘may’). (As mentioned above) the variation in study results between sites deserves mention.

Response: This bullet has been changed to (addition in italics): “Due to the characteristics of the larger study within which this assessment was embedded, there was likely participant selection bias towards infants who were considered unwell. Thus, sensitivity and specificity may be different in other populations, in particular the general neonatal population not restricted to those considered unwell.”

4. Introduction:

- a. Page 5, end of first paragraph – the risk of stillbirth relates to fetal anaemia and hydrops, not jaundice. Suggest change in wording to clarify this.

Response: This has been modified to (addition in italics): “Globally in 2010 an estimated 481,000 newborn infants were at risk of extreme hyperbilirubinemia, and of these, 24% were at risk of neonatal death, 13% for kernicterus and 11% for stillbirth due to severe hemolytic disease manifest in utero as progressive anemia and hypoalbuminemia, leading to hydrops fetalis.”

- b. Page 7, first paragraph – Low and middle income countries are not defined anywhere in the paper. Suggest using a citation for criteria.

Response: The following reference has been added: Serajuddin U, Hamadeh N. New World Bank country classifications by income level: 2020-2021. <https://blogs.worldbank.org/opendata/new-world-bank-country-classifications-income-level-2020-2021> (accessed 18 August 2021).

- c. Same paragraph – suggest replacing “Point-of-care technologies” with “Point-of-care devices”.

Response: Done

5. Methods:

- a. Page 6 1st paragraph – (minor point) presumably a parent provided informed consent, not the infants?

Response: This has been clarified as follows (addition in italics): “...included in the main study if they were <60 days of age and their parent or guardian provided informed consent...”

- b. The comment about “including sample size determination” is a bit vague. It is not clear to me that the 2008 provides a sample size estimation for the current study. More likely, the sample size for the current study was a sample of convenience, with the numbers available determined by the fact that it was nested in the 2003-2005 study. Is this correct?

Response: The mention of sample size refers to the main study. You are correct, that the sample size for this study was a sample of convenience. This has been clarified with the following addition: “...the sample size for the current study was a convenience sample based on enrolments in the main study.”

c. Table 1

- i. the first row switches from the SI units used elsewhere in the paper to mg/dL.

Response: This has been corrected.

- ii. Also, the calibration range for this device (3.3-5.6 mg/dL) needs some further comment by the authors, because it is nowhere near the range of dangerous bilirubin levels. Do the authors have any confidence in its accuracy in the range of bilirubin levels above 20 mg/dL (~340 micromol/L).

Response: We investigated this with the study site PI and his team and determined that the calibration range was stated in error, and has been deleted. The Dhaka Shishu Hospital laboratory is exceptional for a low resource setting. They used standard spectrophotometric methods and standard quality control measures, and routinely standardized their methods for measurement of serum bilirubin levels across the entire range of values encountered in young infants.

- iii. Is there confidence in the calibration ranges of the analysers used in other locations?

Response: Total bilirubin was determined on all admitted infants and all outpatients with any degree of jaundice whenever possible using standard spectrophotometric methods. Although a standardized approach to laboratory quality control for bilirubin measurements across the sites was not implemented as part of the protocol, all participating sites had highly experienced laboratories as they were tertiary care centers that managed thousands of sick neonates and young infants annually. Although we are unable to provide detailed specifications on the specific tests used by each center, given their level of experience and stature as referral centers within their communities, we have full confidence in their laboratory capacity to measure bilirubin with accuracy even when levels were elevated at 20 mg/dL or more.

- iv. “Calorimetric” in row 3 of the table should probably be “colorimetric”

Response: Done

- d. Clinical assessment of jaundice page 8 – “principle” should be “principal”

Response: Done

- e. The change in protocol and partial use of data from two sites deserves some additional explanation, perhaps in a supplementary document.

Response: In the Discussion we noted that “In Kumasi, where skin pigmentation was darkest among the study locations, study investigators were not confident that they could identify jaundice on the face, chest or thighs; as a result, their assessments of jaundice were limited to the sclerae, gums, palms and soles.” In addition, not all sites started at the same time. We refined the data collection forms for the jaundice assessment after the Kumasi site had already completed enrolment. The following text (in italics) was added: Because sites did not start enrolling at the same time, a change in study protocol partway through the study meant that in Kumasi, which was the first site to start and complete enrollment, limited data were available from the face and chest/abdomen in comparable form to the other study sites. In the Discussion, the following text (in italics) was also added: In Kumasi, where skin pigmentation was darkest among the study locations, study investigators were not confident that they could identify jaundice on the face, chest or thighs; thus, in addition to a change in the multi-site protocol (see methods), their assessments of jaundice were limited to the sclerae, gums, palms and soles.

Data from the La Paz study location were restricted to the first half of data collection, since the distribution of serum bilirubin levels in the second half of the study suggested the introduction of a systematic error despite quality control measures, which could not be identified and addressed; thus, data of suspect quality was excluded.

6. Results:

- a. Prevalence of jaundice and hyperbilirubinemia: Given that the strata for the study were 1-6 and 7-50 days, why are results presented for 1-20 days? Was this an a priori decision? If it was post-hoc, is there risk of selective outcome reporting?

Response: The following sentence has been added to Methods (new text in italics): “Selection of this age range was predetermined to address the gap in IMCI guidelines on clinical signs of jaundice in infants aged 1-20 days which signal urgent referral for hospital-level care.’ Note that the vast majority of young infants with jaundice were 1-20 days of age, and were reasonably well balanced by age strata with 1018 infants less than one week of age (days 1-6) (38%) and 1624 infants one week of age or more (days 7-20) (62%); however, the sample size was too small to accommodate separate analysis by strata.

- b. In the same section, it should be clarified that the denominator for the percentages with high bilirubins is those babies who had a bilirubin measured, not all jaundiced infants.

Response: This has been clarified as follows (addition in italics) “Among infants aged 1-20 days with any jaundice, 962 (78%) had a serum bilirubin measurement recorded, and among infants whose bilirubin level was measured, 278 (29%) and 95 (10%) had hyperbilirubinemia $>260 \mu\text{mol/L}$ ($>15 \text{ mg/dL}$) and $>340 \mu\text{mol/L}$ ($>20 \text{ mg/dL}$), respectively.”

- c. Page 17 2nd paragraph; the sentence “For the higher threshold of hyperbilirubinemia... sensitivity was higher and specificity was comparable to slightly lower compared to the findings at the lower serum bilirubin threshold...” would be clearer if the term “comparable to slightly lower” was replaced. Would “similar” be better?

Response: This change has been made.

7. Discussion:

- a. Page 19 1st paragraph; The sentence: “This latter algorithm may be preferable in settings in which prompt referral and access to timely measurement and management of hyperbilirubinemia are lacking or likely to be delayed” may need

improvement. Some explanation of “prompt referral” is needed because the study was done in (and is presumably applicable to) the hospitals where treatment would usually be provided. Is this not the case? Also, if access to measurement and management are not available at all, the threshold used is moot.

Response: The following phrase has been added to the end of this sentence for clarification: “...such as in many LMIC peripheral healthcare settings with minimal equipment and skilled providers where use of the higher sensitivity algorithm may minimise missing patients who can benefit from initiation of treatment while referral is managed.”

- b. The term “first level health facilities” should perhaps be better defined. On page 20, the term “peripheral health clinics” is used. Are these the same thing?

Response: Yes, the term peripheral healthcare facilities/settings is now used consistently.

- c. Page 20 first paragraph; Re; the sentence “However jaundice that deeply involved the trunk or head was also useful clinically (reviewer question – for what?) and recognition of this less distal but more sensitive sign....may be particularly important...”. It seems self-evident that the trunk and head are “less distal” than palms or soles. However, the reason why this makes an algorithm using facial or truncal jaundice “particularly important” is very unclear.

Response: Thank you, this sentence has been clarified as follows (additions in italics): However, jaundice that deeply involved the trunk or head was also useful clinically for identification of patients with hyperbilirubinemia requiring treatment and recognition of this more sensitive sign, as part of the algorithm “any jaundice of the distal extremities or deep jaundice of the trunk or head” may also be important in low resource settings with fewer diagnostic and treatment capabilities.

- d. Is “algorithm” (a process or set of instructions to be followed) the best word? The term “algorithm” throughout the paper seems to be used to mean “detection site(s)”.

Response: We have changed the language to “visual assessment rules”.

8. References:

- a. In references 7, 8, 27 and 28, there is variation in how World Health Organization is abbreviated and some correction is needed.

Response: Ref 7 is a journal which uses particular abbreviation to conform to referencing standards for journal articles. World Health Organization has been spelled out to be consistent across refs 8, 27 and 28.

VERSION 2 – REVIEW

REVIEWER	Liley, Helen The University of Queensland, Mater Research Institute
REVIEW RETURNED	14-Oct-2021
GENERAL COMMENTS	Thanks for addressing the questions and suggestions comprehensively.